# A Comparative Multi-System Approach to Characterizing Bioactivity of Commonly Occurring Chemicals

**DOI:** 10.3390/ijerph19073829

**Published:** 2022-03-23

**Authors:** Brianna N. Rivera, Lindsay B. Wilson, Doo Nam Kim, Paritosh Pande, Kim A. Anderson, Susan C. Tilton, Robyn L. Tanguay

**Affiliations:** 1Department of Environmental and Molecular Toxicology, Oregon State University, Corvallis, OR 97331, USA; brianna.rivera@oregonstate.edu (B.N.R.); lindsay.wilson@oregonstate.edu (L.B.W.); kim.anderson@oregonstate.edu (K.A.A.); susan.tilton@oregonstate.edu (S.C.T.); 2Pacific Northwest National Laboratory, Biological Sciences Division, Richland, WA 99354, USA; doonam.kim@pnnl.gov (D.N.K.); paritosh.pande@pnnl.gov (P.P.)

**Keywords:** alternative toxicological models, high-throughput screening, normal human bronchial epithelial cells, zebrafish, passive sampling

## Abstract

A 2019 retrospective study analyzed wristband personal samplers from fourteen different communities across three different continents for over 1530 organic chemicals. Investigators identified fourteen chemicals (G14) detected in over 50% of personal samplers. The G14 represent a group of chemicals that individuals are commonly exposed to, and are mainly associated with consumer products including plasticizers, fragrances, flame retardants, and pesticides. The high frequency of exposure to these chemicals raises questions of their potential adverse human health effects. Additionally, the possibility of exposure to mixtures of these chemicals is likely due to their co-occurrence; thus, the potential for mixtures to induce differential bioactivity warrants further investigation. This study describes a novel approach to broadly evaluate the hazards of personal chemical exposures by coupling data from personal sampling devices with high-throughput bioactivity screenings using in vitro and non-mammalian in vivo models. To account for species and sensitivity differences, screening was conducted using primary normal human bronchial epithelial (NHBE) cells and early life-stage zebrafish. Mixtures of the G14 and most potent G14 chemicals were created to assess potential mixture effects. Chemical bioactivity was dependent on the model system, with five and eleven chemicals deemed bioactive in NHBE and zebrafish, respectively, supporting the use of a multi-system approach for bioactivity testing and highlighting sensitivity differences between the models. In both NHBE and zebrafish, mixture effects were observed when screening mixtures of the most potent chemicals. Observations of BMC-based mixtures in NHBE (NHBE BMC Mix) and zebrafish (ZF BMC Mix) suggested antagonistic effects. In this study, consumer product-related chemicals were prioritized for bioactivity screening using personal exposure data. High-throughput high-content screening was utilized to assess the chemical bioactivity and mixture effects of the most potent chemicals.

## 1. Introduction

### 1.1. Discovery of Common Chemical Exposures

Individuals are exposed to complex mixtures of chemicals from multiple chemical classes on a daily basis. Exposure to complex mixtures can occur from numerous sources, including but not limited to consumer products, food contact chemicals, and air and water pollution [1]. To better understand real-world exposure to complex mixtures, in 2019 Dixon et al. investigated chemical exposures using personal passive sampling devices in the form of silicone wristbands [2]. The same researchers conducted a retrospective study investigating personal chemical exposures from fourteen unique communities for a total of 262 wristbands. Wristbands were analyzed, using an in-house analytical method on an Agilent 7890A GC with a 5975C MS detector [3] for the presence of over 1530 organic chemicals commonly found in personal care products, household products, industrial and agricultural processes, or derived from natural sources. Results from this study highlighted the uniqueness of personal chemical exposures, with no two wristbands having the same chemical exposure profile. Importantly, researchers identified fourteen chemicals in over 50% of all wristbands, which will be referred to as the Global 14 chemicals (G14). These chemicals are commonly found in consumer products and are primarily used as plasticizers, fragrances, flame retardants, or pesticides (Table 1). While wristband passive sampling devices cannot provide definitive data on the sources of chemical exposures, the chemical structure classification and functional uses observed can be indicative of common sources of exposure, such as fragrances and flame retardant compounds being associated with consumer products commonly found around the home.

### 1.2. Exposure to G14 Chemicals

Six of the G14 chemicals are phthalates. Phthalates are a class of compounds used in a variety of consumer products including clothing, cosmetics, pharmaceuticals, food packaging, and hundreds more. They are largely used as plasticizers to add flexibility and durability to plastics; several phthalates are primarily used as solvents and may be found in personal care products to add lubrication or carry fragrance [5]. Greater use of personal care products such as makeup and sunscreen is associated with increased exposure to phthalates, as indicated by a greater concentration of urinary phthalate metabolites [6].

HHCB, AHTN, lilial, and BS are all associated with fragrance additives in personal care products. A 2015 study assessed daily usage of scented personal care products and their formulation contents of twelve synthetic musks, identifying HHCB and AHTN as accounting for a total of 45% of the daily dermal exposure in adults [7]. DEET is the active ingredient in many pest repellents and may be applied directly to the skin. It has been estimated that up to a third of the US population uses DEET-containing products each year [8]. Triphenyl phosphate, or TPP, is used as a plasticizer and flame retardant in a variety of consumer products. Occupational exposures to TPP have been reported in industrial settings, nail salons, and amongst aircraft maintenance technicians [9,10,11].

### 1.3. Importance of Studying Chemical Mixtures

Addressing research gaps for chemical mixture assessment has been identified as a priority by regulatory agencies in both the European Union and the United States [1,12]. These agencies highlight the importance of considering chemical mixtures for both exposure and hazard assessments, as this represents a more realistic exposure scenario [13,14]. However, investigations of chemical exposures and their associated hazards has historically been conducted on a chemical-by-chemical basis. Mixtures can consist of chemicals with similar or dissimilar mechanisms of action, which may influence biological response [13,15,16,17]. When compared to their individual components, mixtures have been shown to demonstrate differences in their toxicity profiles. Biological responses to mixtures are broadly grouped into three categories: additivity, synergy, or antagonism [13]. Generally, mixture effects are tested with the assumption that chemicals are eliciting toxicity in an additive manner. Testing dose or concentration additivity is typically applied to groups of chemicals that are believed to have the same mechanism of action, and the mixture effects can be predicted by adding the concentrations of the chemicals found in the mixture [13,16,17,18]. If the exact mechanisms of all components in a mixture are unclear, it is suggested that the concentration addition approach be used, as it provides a more conservative estimate [19,20,21].

### 1.4. Use of Two Alternative Models to Assess Chemical Bioactivity

Due to scientific need and public concerns over animal welfare, there has been a push to move away from traditional rodent testing for chemical bioactivity screening based on the 3Rs principles (Reduction, Replacement, and Refinement) [22]. While traditional methods may present certain inherent advantages for measuring specific toxicity endpoints, advances in the development of alternative models have made it possible to detect chemical bioactivity of large libraries of chemicals in a high-throughput manner [22,23,24,25,26].

The utility of in vitro methods for chemical prioritization has largely been demonstrated through the U.S. Environmental Protection Agency’s ToxCast program [22,26]. Primary human cell culture is representative of the human response to chemical exposures, and more predictive of toxicological effects to human target tissues [24,25]. In this study we used primary normal human bronchial epithelium (NHBE) as an in vitro model for high-throughput bioactivity testing of common chemical exposures identified in Dixon et al., 2019 [2]. Personal exposures to semi-volatile organic chemicals (SVOCs) were investigated. SVOCs can be found in the vapor or particle-bound phase, making inhalation an important route of exposure [27]. Primary NHBE were selected to serve as a representative in vitro system of human response for inhalation exposure and predictive toxicological effects.

The zebrafish (Danio rerio) is a well-established model for biomedical and toxicological sciences. At early life-stages, the zebrafish is a superb model for developmental toxicity testing and high-throughput bioactivity screening of chemicals [28,29,30,31]. Zebrafish and humans have remarkably high genetic similarity and share many of the same internal organs, making the model highly relevant to human health research [32]. Additionally, zebrafish develop rapidly and transparently, allowing for observation of organogenesis and teratogenic effects of chemicals in just five days [33]. This ease of use and relevance to human health make the zebrafish an ideal model for toxicity testing, with the major advantage of capturing whole-animal biological complexity that is unavailable with in vitro and in silico screening.

### 1.5. Bridging the Gap between Exposure Science and Toxicology

To characterize the risk of chemicals to human health and the environment, understanding the frequency and concentrations of chemicals and their potential to interact with biology are both equally necessary. A chemical may not be considered hazardous and worthy of regulation based on known exposure alone. Similarly, a chemical with known bioactivity and little exposure to organisms may not present a significant hazard. By utilizing curated data on known chemical exposures, researchers can identify the chemicals humans are most likely to come into contact with, allowing us to prioritize commonly occurring chemicals for hazard assessment [34,35]. Approaches utilizing in vitro methods, machine learning, and high-throughput screening in lower vertebrates have given us an opportunity to bridge the gap between exposure science and toxicology [36,37,38]. In this study, we present a novel approach utilizing known human exposure data to inform comparative multi-system high-throughput bioactivity screening. By taking a multi-system approach using both in vitro NHBE cells and in vivo zebrafish models for toxicity testing, we aim to utilize two commonly used high-throughput methods for chemical screening, accounting for species differences, sensitivity differences, and whole-animal complexity. This approach is intended to prioritize chemicals that may be further investigated to link specific biological outcomes to chemical exposures.

## 2. Materials and Methods

### 2.1. Chemicals and Reagents

Neat chemical stocks were purchased and dissolved in ACS grade n-hexane (Thermo Fisher Scientific, Waltham, MA, USA). HHCB was created at 10 mM due to limits of solubility, and other stock solutions were created at 20 mM. Stock solutions in n-hexane were stored at 4 °C until use. Prior to use, stock solutions were solvent exchanged into ACS-grade dimethyl sulfoxide (DMSO) (Thermo Fisher Scientific, Waltham, MA, USA) and stored at room temperature. Table 2 details the compound category, CAS registry numbers, supplier information, and purity.

### 2.2. Mixture Calculations and Concentration Selection

Chemicals were selected for this study based on frequency of detection in Dixon et al., 2019. Selected exposure concentrations were intended to capture the range of any observed effects for each respective model system in order to compare the potency of bioactive chemicals. Exposure concentrations for individual chemicals were capped based on the maximum effects observed in each model system. Maximum effects were determined based on previously established methods or based on levels in which response saturation was observed. Concentrations were capped for definitive testing at 200 for NHBE and 100 µM for zebrafish.

An equimolar mixture of all fourteen components was created to assess the toxicity of the G14 chemicals identified in Dixon et al., 2019 [2]. The fourteen-component equimolar mixture (G14 Mix) was created using stock solutions in n-hexane (Thermo Fisher Scientific, Waltham, MA, USA) with each component at a final concentration of 1 mM due to limits of solubility. Prior to use, the G14 Mix was solvent exchanged into ACS-grade DMSO. All equimolar mixture exposure concentrations are listed as the concentration of the sum of the components, e.g., the G14 Mix mixture at 140 µM exposure concentration is made up of fourteen chemicals, each at a 10 µM concentration within the whole mixture.

Bioactivity-based mixtures were created upon observing the bioactivity patterns of individual chemicals across zebrafish and NHBE models. For NHBE exposures, mixtures were formed from the individual stock solutions listed in Table 1 and diluted to target concentrations using ACS-grade DMSO and stored at room temperature. For zebrafish mixtures, chemicals were added individually directly to plated embryos with gentle shaking to ensure thorough mixing.

An equimolar sub mixture containing the top three most potent compounds detected by each model was screened on its respective platform. For the NHBE mixture (NHBE Equi-Mix), each of the most potent chemicals in NHBE cells was used to create a three-component equimolar mixture comprised of AHTN, BHT, and HHCB at a maximum stock concentration of 15 mM with each individual component at a maximum exposure concentration of 100 µM, for a total of 300 µM. An NHBE BMC-based mixture was created to account for differences in potency between chemicals, and contained the same three chemicals mentioned above at a maximum concentration of the BMC_50_ of each chemical. Each mixture concentration was reported as the sum of their components. For the zebrafish model, an all-phthalate equimolar mixture (ZF Equi-Mix) was formed by addition of BBP, DBP, and DIBP directly to plated embryos with each at a maximum nominal water concentration of 2.5 µM, for a total concentration of 7.5 µM.A zebrafish BMC-based mixture comprised of the same three chemicals as its equimolar counterpart was formed with a maximum concentration as the BMC_10_ of each chemical.

Concentrations of BMC mixtures were reported as the sum of their components. Decreasing concentrations for each component were determined as fractions of their BMC value; thus, the ratio of mixture components was equal for every concentration. All exposure concentrations for each mixture can be found in Table 3 and Table 4. Concentrations of individual chemicals in the NHBE and ZF BMC Mix can be found in Appendix A. For each model system, DMSO was used as a vehicle control.

### 2.3. NHBE Bioactivity Screening

#### 2.3.1. Cell Culture Conditions and Exposures

Normal Human Bronchial Epithelium cells (NHBE) (passage 4; Lonza, Walkersville, MD, USA) were expanded in a T75 culture flask in Pneumacult-Ex Plus media (STEMCELL Technologies, Vancouver, BC, Canada) until 90% confluency was reached. Cells were then trypsinized and plated in black-walled 96-well plates at a density of 1.3 × 10^4^ cells/well. Cells were maintained in 200 µL of Pneumacult-Ex Plus media at 37 °C and 5% CO_2_ until 90% confluency was reached. Chemical stocks were diluted to 2% DMSO (*v*/*v*) with Pneumacult-Ex media (STEMCELL Technologies, Vancouver, Canada). Cells were exposed for 24 h at 37 °C and 5% CO_2_. Plates were run in duplicate to account for between plate variability (*n* = 6/concentration). Initial range-finding experiments were conducted with a four-point concentration response ranging from 20 to 400 µM. Final concentrations of individual chemicals ranged from 10 to 200 µM with a six-point concentration response. Concentrations for the NHBE Equi-Mix and G14 mix ranged from 30 to 300 µM and 28 to 280 µM, respectively, with a five-point concentration response. The NHBE BMC mixture ranged from final nominal concentrations of 11.5 to 230.4 µM, with a seven-point concentration response. Treatment-specific concentrations can be found in Table 3.

#### 2.3.2. Lactate Dehydrogenase Assay

Lactate Dehydrogenase (LDH) leakage was measured in cell media as an indicator of cytotoxicity from chemical treatments using Cyquant LDH Kit (Thermo Fischer Scientific, Waltham, MA, USA). Menadione (200 µM), which has shown to induce significant cytotoxicity in our cells, was used as a positive control and cells incubated in cell media, and 2% DMSO served as a vehicle control. After the 24-h exposure period, equal volumes of cell media and LDH reagent were transferred to a fresh 96-well plate and incubated away from light for 30 min. An equal volume of the stop solution was added, and absorbance was read at 490 nm and background at 680 nm using a Synergy HTX plate Bio Tek plate reader (Winooski, VT, USA). Cytotoxicity was calculated by subtracting background from absorbance.

#### 2.3.3. Cell Titer Glo Assay

Cell viability was measured using a Promega CellTiter-Glo Luminescent Cell Viability Assay (Madison, WI, USA). Cell viability is typically measured based on quantification of ATP, which serves as an indicator of metabolically active cells. Media with 2% DMSO served as the vehicle control and menadione served as the positive control. After the 24-h exposure period, the plate was brought to room temperature and an equal volume of the CellTiter-Glo reagent was added to the black-walled 96-well plate. The plate was then protected from light and placed on an orbital shaker at 10 rpm for 15 min. Full-spectrum luminescence was then read using a Synergy HTX plate Bio Tek plate reader (Winooski, VT, USA).

#### 2.3.4. 2′,7′-Dichlorofluorescin Diacetate (DCFDA) Assay

Reactive oxygen species were measured using DCFDA (Sigma-Aldrich, St. Louis, MO, USA). Neat stock was purchased and diluted in ACS-grade DMSO. After cells were incubated with target compounds, vehicle control, and positive control for 24 h, the dosing solutions were removed and 20 µM DCFDA diluted with HBSS to 2% DMSO was pipetted onto the cells. The plate was then immediately read using a Synergy HTX plate Bio Tek plate reader (Winooski, VT, USA) at 485/528 excitation and emission.

#### 2.3.5. NHBE Statistics

Treatment effects for each in vitro assay were investigated using values normalized to vehicle control. Treatment effects relative to vehicle control were then investigated using one-way ANOVA with Dunnett’s post hoc test. Pairwise comparisons of respective concentrations between individual chemicals and mixtures were investigated using the Tukey–Kramer honestly significant difference test. A significance level was defined with a *p*-value cutoff of 0.05 for both analyses.

### 2.4. Zebrafish Bioactivity Screening

#### 2.4.1. Zebrafish Husbandry and Exposures

In accordance with the Institutional Animal Care and Use Committee protocols at Oregon State University (IACUC-2021-0166 and 2021-0227), specific pathogen-free wild type 5D zebrafish (Danio rerio) [39] were reared at the Sinnhuber Aquatic Research Laboratory (SARL). Fish were housed in brood stock (50- or 100-gallon) tanks on a recirculating water system kept at 28 ± 1 °C under a 14:10 h light–dark cycle. To maintain optimal water conditions, water was supplemented with Instant Ocean salts (Spectrum Brands, Blacksburg, VA, USA) and sodium bicarbonate as needed to maintain pH 7.4. Fish were fed twice daily with Gemma Micro (Skretting, Inc., Fontaine Les Vervins, France) [40].

On the day of exposure, embryos were collected using an internal embryo collection apparatus, sorted by similar developmental stage, and kept in E2 embryo medium (EM) consisting of 15 mM NaCl, 0.5 mM KCl, 1 mM CaCl_2_, 1 mM MgSO_4_, 0.15 mM KH_2_PO_4_, 0.05 mM Na_2_HPO_4_, and 0.7 mM NaHCO3 buffered with 1 M NaOH to pH 7.2 [41]. Embryos were held in a temperature-controlled incubator at 28 ± 1 °C until dechorionation.

At 4 h post-fertilization (hpf), embryos were enzymatically dechorionated using a custom-made automated dechorionator, previously described in [42]. Dechorionated embryos were screened for enzymatic or mechanical damage under a dissecting microscope, then undamaged embryos were robotically loaded into 96-well round-bottom plates prefilled with 100 µL EM.

Chemicals were dispensed into 96-well plates pre-loaded with embryos and EM using an HP D300 or D300e Digital Dispenser, then immediately sealed using an Eppendorf 5390 heat sealer with pressure-sensitive silicone adhesive backed polyolefin plastic PCR film (Thermaseal RTS). Plates were incubated at 28 ± 1 °C overnight on an orbital shaker at 235 RPM under dark conditions.

For each chemical, embryos were statically exposed to initial range-finding nominal water concentrations of 0, 1, 2.54, 6.45, 16.4, 35, 74.8, and 100 µM (1 plate, *n* = 12 for each concentration) beginning at 6 hpf and continuing until 120 hpf. Exposure solutions were normalized to 0.64% by volume of DMSO. Definitive testing concentrations were selected upon screening at five days post-fertilization (dpf) to capture the full range of effects from 0% to 100% bioactivity. In the instance that less than 100% bioactivity was observed under range-finding, test concentrations were capped at 100 µM. Final test concentrations for each chemical and mixture can be found in Table 4.

#### 2.4.2. Zebrafish Morphology Screening

Zebrafish remained under static exposure from 6–120 hpf to cover the period of early development through organogenesis. Fish were screened for a total of thirteen morphological endpoints at 24 hpf and 120 hpf by visual assessment under a dissecting microscope. Table 5 lists the morphological endpoints assessed at each timepoint. To assess effects on morphology, percent incidence of abnormalities for each endpoint was calculated across three test plates (*n* = 36 for each concentration) for each chemical or mixture. The percent incidence of any observed morphological effect was calculated and reported as “any effect”. Images of all measured endpoints can be found at https://github.com/Tanguay-Lab/Bioinformatic_and_Toxicological_Resources/tree/main/Files/Zebrafish_Phenotype_Atlas (accessed on 21 March 2022); see Appendix A regarding morphological endpoint binning. Prior to analysis, data were quality controlled for incidence of background malformations in DMSO vehicle controls, with a required threshold of >80% normal fish with no mortality or malformations at 120 hpf. All zebrafish screening data presented here met this criterion. The incidence of each malformation at each concentration was compared to that of the DMSO vehicle control using Fisher’s Exact test (*p* < 0.05), as described previously [43,44].

#### 2.4.3. Embryonic and Larval Photomotor Response Assays

An embryonic photomotor response (EPR) assay was conducted at 24 hpf prior to morphological assessment, taking care to not expose the test plates to visible light prior to the assay [45]. Briefly, EPR videos were captured only with infrared lighting, with a stimulus consisting of two 1 s pulses of white visible light at 30 and 40 s after video recording began. The nine seconds prior to the first pulse were considered the “background” (B) period; the nine seconds immediately after the first pulse were considered the “excitatory” (E) period; the nine seconds following the second pulse were considered the “refractory” (R) period. Data associated with dead or developmentally delayed embryos were removed prior to analysis. Response in treatment groups was compared to that of controls based on movement index using the Kolmogorov–Smirnov test (Bonferroni corrected *p*-value of 0.05) [45]. In this assay, embryos may exhibit normal, hypo-, or hyperactivity relative to the on-plate control animals at any of the three timepoints, indicating chemical-induced effects on non-visual photomotor development [46].

A larval photomotor response (LPR) assay was conducted at 120 hpf, as previously described [47]. Briefly, test plates were placed into ZebraBox behavioral analysis chambers (Viewpoint Life Sciences) and larval movement was tracked with ZebraLab motion analysis software across four cycles of 3 min light and 3 min dark for a total of 24 min, with the first three cycles considered an acclimation period and the final cycle considered the test period. The distance moved by each larva was integrated over 6 s binning periods, and total distance moved was plotted against time. Differential entropy was modeled and response of treatment groups was compared to that of controls using area under the curve ratios and Kolmogorov–Smirnov test [47]. Data associated with dead or malformed larvae were removed prior to analysis, with data reported for treatments with at least 70% normal larvae. For this assay, embryos may exhibit normal, hypo-, or hyperactivity relative to control at the light and/or dark periods, indicating chemical-induced effects on non-visual and/or visual photomotor development.

For all zebrafish bioactivity assessments, data were uploaded under a unique well-plate barcode into a custom LIMS database, the Zebrafish Acquisition and Analysis Program (ZAAP), and analyzed using custom R scripts that were executed in the LIMS background [48]. For photomotor response data, a chemical or mixture was considered bioactive if at least three consecutive concentrations significantly elicited the same behavioral response.

### 2.5. Correlation Matrix of Real-World Exposures

Co-occurrence of G14 chemicals is likely due to overlapping functional use of these chemicals and to the majority being found in consumer products. In order to understand which of these chemicals are most likely to occur together, real-world exposure concentrations of G14 chemicals detected in Dixon et al., 2019, were obtained. Correlations between G14 chemical concentrations normalized by deployment time were investigated using “ggcorrplot” in R [49,50]. Significant correlations were defined with a *p*-value cut-off of 0.01. Plots were generated using a correlation matrix and a matrix of correlation *p*-values. Only pairs of chemicals with *p* < 0.01 were shown and labeled with their correlation coefficients. Correlation matrices for all G14 chemicals can be found in Appendix A.

### 2.6. Benchmark Concentration Modeling and Mixture Interaction Assessment

Curve fittings for concentration response curves were conducted using eight dose-response models (logistic, gamma, weibull, probit, log-logistic, log-probit, multistage, and quantal linear), as described in Gosline et al., 2021 [51]. Following EPA guidance, a data adequacy assessment and best model selection were carried out and the models of best fit were used to predict benchmark concentrations [52]. A concentration addition equation was applied for the investigation of mixture interactions, which the U.S. EPA recognizes as a default model based on its propensity to provide more conservative estimates [15,52]; a mixture interaction index greater than 1 is indicative of antagonism, and a mixture interaction index of less than 1 is indicative of synergism. If the mixture interaction index is equal to 1, then the assumption of additivity is met: (1)$$\overset{c}{\underset{i=1}{\sum }}{\frac{x_{i}}{E}}_{i}=1$$
where *i* corresponds to the individual chemicals in the mixture, *x_i_* is the concentration of chemical *i* at the BMC of the mixture, and *E_i_* is the BMC_50_ of the chemical *i* [14].

## 3. Results

### 3.1. G14 Mixture and Individual Chemical Screening

To identify toxicity profiles of G14 chemicals as a mixture (G14 Mix) and individually, three endpoints in NHBE related to oxidative stress, cytotoxicity, and cell viability were assessed (Appendix A). The G14 mix did not induce significant responses for oxidative stress or cytotoxicity endpoints. However, significant effects were seen for cell viability, with a lowest effect level (LEL) starting at 28 µM (Figure 1; Appendix A). For the individual chemical screening, no chemicals induced oxidative stress. HHCB, AHTN, BHT, and DIBP caused significant cytotoxicity. Significant effects were seen at concentrations equal to or less than 100 µM, with the exception of DIBP. HHCB, AHTN, BHT, DIBP, and TPP caused significant reductions in cell viability relative to control at concentrations less than 100 µM, with the exception of DIBP and TPP (Figure 1; Appendix A). Cell viability was identified as the most sensitive endpoint and will henceforth be discussed as the primary indicator of cell health. TPP and DIBP were not further investigated due a significant response only being observed at the highest tested concentration. HHCB, AHTN, and BHT were identified as the most potent compounds, having multiple concentrations significantly different from control for cell viability and LELs ranging from 10 to 50 µM. Concentration–response curves and bar plots of the most potent chemicals for cell viability and cytotoxicity can be found in Appendix A.

For bioactivity screening in the developmental zebrafish model, fish were screened for thirteen morphological effects including mortality as well as two behavioral endpoints in response to the Global 14 chemicals, both individually and in an equimolar mixture (G14 Mix). The G14 Mix induced significant morphological effects, including craniofacial axis edemas, lower trunk and brain malformations, mortality, and altered touch response, with an LEL of 56 µM for any effect (Figure 1). The G14 Mix induced effects in every endpoint observed in response to its individual components except for musculature (“musc/circ”), which was only observed in response to BS. Because BS was the only chemical to induce effects in this endpoint, it is likely that the dilution of BS in the mixture was enough that it did not reach a sufficient concentration to induce the effects seen when fish were exposed to BS individually.

Nine of the fourteen chemicals induced morphological effects individually, with the effects of BBP, DBP, DIBP, and BS dominated by mortality at 120 hpf (Figure 1). HHCB, AHTN, lilial, and TPP toxicity profiles were dominated by edemas, cranial malformations, and bent body axis. Additionally, HHCB, BS, and TPP induced a significant lack of response to touch. Percent response data for each endpoint for each chemical can be found in Appendix A. Concentration–response curves for any morphological effect can be found in Appendix A.

To increase the sensitivity of the in vivo assay, zebrafish were screened at 24 and 120 hpf for photomotor response indicative of early life-stage behavior effects (Figure 1). Eleven individual chemicals and all mixtures induced abnormal EPR activity, including two, DEET and DEP, which only exhibited EPR effects with no gross morphological effects at any concentration at 120 hpf, reflecting the high sensitivity of this assay. No chemical elicited LPR effects at 120 hpf. This rigorous LPR assay assessment criteria requires 70% normal larvae (no mortality or malformations) at 120 hpf. Because many of the G14 chemicals produced morphological effects at low test concentrations by 120 hpf, we did not identify an LPR effect for any of the chemicals. Summarized EPR data can be found in Appendix A and LPR data in Appendix A.

### 3.2. Investigation of Mixture Interactions

To investigate potential mixture interactions of the G14 mix, toxicity profiles of bioactive components to which each model was particularly sensitive were assessed in equimolar and BMC-based mixtures (Table 6).

#### 3.2.1. NHBE Mixtures

All concentrations in NHBE Equi-Mix (30 to 300 µM) produced significant responses compared to control for cell viability. For concentrations of individual components significantly different from control, pair-wise comparisons were made at respective concentrations in NHBE Equi-Mix. AHTN and HHCB were more potent than NHBE Equi-Mix for concentrations ranging from 25 to 100 µM. BHT was more potent than NHBE Equi-Mix at 75 and 100 µM (Appendix A). Concentration response relationships for the effects of NHBE Equi-Mix and NHBE BMC Mix on cell viability compared to the individual components were modeled using the best fit regression curves and BMC_10_ and BMC_50_ were predicted (Figure 2; Table 7). NHBE Equi-Mix and NHBE BMC mix had BMC_50_ values of 53.9 µM and 94.9 µM, respectively. HHCB, AHTN, and BHT had BMC_50_ values of 42.2, 84.9, and 103 µM, respectively. Investigation of chemical interactions suggested that chemicals in NHBE Equi-Mix were less than additive, or synergistic, with an interaction index of 0.93. In NHBE the results were suggestive of antagonism, with an interaction index of 1.22.

#### 3.2.2. Zebrafish Mixtures

The zebrafish model was particularly sensitive to BBP, DBP, and DIBP. Because these belong to the same chemical class (phthalates) and induced highly similar bioactivity profiles, an equimolar mixture of active phthalates (referred to as the ZF Equi-Mix) was screened in zebrafish, with a maximum concentration of 2.5 µM each and top mixture concentration of 7.5 µM. Concentration selection for mixture exposures were based on mortality and morphology findings from screening of their individual components. The percent incidence of any morphological effect induced by the ZF Equi-Mix and its individual components was modeled across concentrations compared to control animals using the best fit regression model, and BMC_10_ and BMC_50_ were predicted. Individually, BBP, DBP, and DIBP induced bioactivity, with BMC_50_ values of 4.28, 4.27, and 5.32 µM, respectively, while the ZF Equi-Mix mix had a BMC_50_ of 5.36 µM (Table 7). When testing the assumption of additivity to assess potential mixture effects the results suggested antagonism, with an interaction index of 1.17 for the equimolar ZF Equi-Mix. Interestingly, the ZF Equi-Mix did not induce significant mortality at 24 hpf, while each of its components did (Figure 3). In zebrafish, investigation of mixture effects for the BMC Mix suggested antagonism, with an interaction index of 1.40.

## 4. Discussion

### 4.1. Model-Specific Differences in Bioactive Chemical Detection

Model-specific differences were observed using a multi-system approach to chemical bioactivity screening. Five chemicals were detected as bioactive in NHBE and eleven in the zebrafish model, with AHTN, HHCB, DIBP, and TPP in common. The increased sensitivity to bioactive chemicals in zebrafish can be explained by a number of factors, including dosimetry differences between the models, differential metabolism, and interactions between organ systems. It is likely that the sheer number of developmental processes occurring during the first few days of zebrafish development provide many opportunities for a chemical to interfere and disrupt normal development, while NHBE cultures have a more limited number of potential chemical targets.

The three most potent chemicals in NHBE were AHTN, HHCB, and BHT. Interestingly, HHCB, the most potent chemical in NHBE, was the second most-observed chemical in Dixon et al. (2019) with a detection value of 94% [2]. Studies using in vitro models have identified reductions in cell viability driven by apoptosis in response to exposure to HHCB, AHTN, and BHT [53,54]. The primary health concerns around HHCB and AHTN are mostly focused on their endocrine disruption potential, with particular concern regarding effects on aquatic organisms [55,56]. BHT has been reported to affect allergic diseases and rhinitis, endocrine disruption potential, and aquatic toxicity [57,58,59,60].

The only chemical which was detected as bioactive in NHBE which did not show bioactivity in the zebrafish model was BHT. Previous studies in developmental zebrafish have demonstrated conflicting results regarding BHT-induced toxicity. Multiple studies suggest BHT is toxic, with a 96-h LC50 ranging from ~20–60 µM and larval behavior being affected at as low as 1 µM [57,61]; however, a separate study did not find significant lethality up to 200 µM, though physical malformations were observed below 100 µM [58]. The inconsistency between our findings and those of previous studies may be attributed to variable exposure conditions, including chorion status and exposure in pools versus individual wells, both of which can affect the actual internal dose of a chemical [62,63,64]. Additionally, while the developmental zebrafish model is highly sensitive to detecting bioactive chemicals, whole-animal bioactivity screening does not provide mechanistic information about a chemical. Discrepancies between detection of bioactivity between in vitro and in vivo models may be attributed to differences in xenobiotic metabolism and/or the developmental stage during exposure.

The zebrafish model exhibited particular sensitivity to phthalates. The three most bioactive compounds identified from zebrafish bioactivity screening were DBP, DIBP, and BBP, which are all phthalates with BMC_50_ values below 5.5 µM. The other phthalates, DEP, DNP, and DEHP, did not induce any morphological effects in zebrafish up to 100 µM. These findings are consistent with previous screening efforts regarding phthalates with respect to gross morphological effects, although some studies have identified effects of these compounds on biomarkers related to neurotoxicity and endocrine disruption such as increased vitellogenin and triiodothyronine (T3) levels [65,66]. Several of the phthalates tested have been associated with “phthalate syndrome”, a cluster of developmental reproductive effects including lowered fetal testicular testosterone and physical genital abnormalities, namely shortened anogenital distance [67,68,69,70]. One reason we may not have identified certain phthalates, such as DEHP and DEP, as morphologically active in the zebrafish model is the absence of target tissues, specifically external reproductive organs where most mammalian effects are observed [71]. Additionally, the exposures performed in this study ended at 120 hpf, prior to key reproductive events such as sex differentiation and production of germ cells. The lack of bioactivity we observed with exposure to DEHP and DEP may be due to the target tissues not yet being present at 5 dpf, or the effects may simply be unobservable until later in life [72].

In addition to the active phthalates, TPP, BS, HHCB, AHTN, BP, and lilial induced significant malformations in developing zebrafish. Each of these compounds has been associated with biomarkers of endocrine disruption in laboratory studies [56,73,74,75,76,77,78,79]. Disruption of normal hormone synthesis and signaling can have widespread effects in whole animals, including defects or dysfunction of the cardiovascular, nervous, and immune systems [80,81,82,83,84,85]. It is evident that for many known and suspected endocrine-disrupting compounds, the zebrafish model is sensitive enough to detect bioactivity even in the absence of external reproductive organs. Importantly, each of these compounds may induce toxicity via divergent mechanisms, whether through endocrine disruption or otherwise. Investigation into this is beyond the scope of this study; nevertheless, this common determination of G14 chemicals as potential EDCs and their widespread bioactivity in the zebrafish model should be noted.

An embryonic photomotor assay identified bioactive effects of eleven chemicals, including DEET and DEP, which induced no malformations or mortality, highlighting the sensitivity of this assay. At 5 dpf, LPR did not identify any chemical as bioactive. In order to avoid confounding behavioral effects with physical malformations, only non-malformed fish and only concentrations which induce <30% mortality and malformations at 5 dpf were considered for LPR analysis. Nine of the G14 chemicals induced morphological effects at low concentrations, meaning that while LPR may have been altered by these chemicals we could not be certain that the effects were due to actual neurodevelopmental deficits. Exposing fish to these same chemicals at a lower concentration range may uncover LPR effects.

Two chemicals, DEET and DEP, induced only EPR effects, with insignificant malformations and no LPR response. While both EPR and LPR are behavioral assays, they are not necessarily indicative of the same biological response; thus, it cannot be assumed that if a chemical induces EPR effects, it will induce LPR effects. EPR is measured at 24 hpf, before the eyes develop, and movement in response to light is largely controlled by photomotor cells in the hindbrain which mediate response to light stimulus [46]. LPR is measured at 120 hpf, when the eyes are formed and musculature is more established, meaning that the LPR response can be indicative of developmental disruptions of the nervous system, musculature, vision, and more [45,86,87]. Both assays can indicate the potential for a chemical to interfere with normal development, adding to the weight of evidence of a chemical’s observed bioactivity; however, they are not always correlated with one another.

### 4.2. Investigation of Mixture Effects

Due to the common use and occurrence of the G14 chemicals there exists a large body of literature on the biological effects of some of the individual chemicals, although published data on bioactivity of relevant mixtures of these compounds is minimal. Mixture interactions of bioactive chemicals were further investigated in each model system. Concentration–response curves of the NHBE Equi-Mix and ZF Equi-Mix were suggestive of mixture effects with a much lower BMC_50_ compared to the individual components. Comparison of concentration–response curves for the G14 mixture in NHBE were suggestive of mixture effects, with a BMC_50_ of 47.7 µM compared to 53.9 µM in the NHBE Equi-Mix. The lower BMC_50_ of the G14 compared to NHBE Equi-Mix may imply that components in the NHBE Equi-Mix are not the sole driver of observed toxicity from exposure to the G14 mixture, and that there are other chemical components that may be interacting to elicit toxicity in NHBE.

Investigations of mixture interactions using the concentration addition model resulted in different outcomes for the NHBE Equi-Mix and the NHBE BMC Mix. Results for the NHBE Equi-Mix suggested synergism, while those for the NHBE BMC Mix suggested antagonism, with respective interaction indexes of 0.93 and 1.22. The exact mechanisms of bioactive chemicals in each model system are unclear; our use of the concentration addition approach may be a conservative estimate of the actual mixture effects if these chemicals are acting dissimilarly [19,20]. Zebrafish screening identified three phthalates as the most potent bioactive chemicals. When screened in both equimolar and BMC-based mixtures, assumption of additivity tests suggested antagonism. Interestingly, when exposed in equimolar mixture the specific endpoints driving toxicity differed between individual compounds and the mixture. Figure 4 displays the percent incidence of each morphological endpoint measured for both mixtures and their components. Specifically, each active phthalate’s bioactivity was driven by mortality, with DBP bioactivity driven almost entirely by mortality at 24 hpf. Both ZF Equi-Mix and ZF BMC Mix produced similar responses across morphological endpoints, with neither mixture inducing significant mortality at 24 hpf at any concentration. The ZF BMC Mix concentrations were based on the BMC_50_ for each individual chemical, which was similar for BBP, DBP, and DIBP, meaning the total mixture concentration range was similar for these two mixtures.

Differences in mixture potencies were observed based on the mixture formation approach in both model systems. Equimolar mixtures, which did not account for differences in potency between individual chemicals, appeared to be driven by the most potent chemical. In both model systems, the equimolar mixture had a lower BMC_50_ compared to the BMC-anchored mixture. This difference was more pronounced in NHBE, with a BMC_50_ of 54.0 and 94.9 µM for NHBE Equi-Mix and BMC Mix, respectively. In zebrafish, slight differences were observed, with BMC_50_ values of 5.36 and 6.41µM for ZF Equi-Mix and BMC Mix, respectively. The degree of difference in BMC_50_ values between the mixture formation approaches in each model system is believed to be driven by differences in chemical potency. In NHBE, a wider range of BMC_50_ values was observed, ranging from 42.2 µM to 103 µM. In zebrafish, BMC_50_ values had a much smaller range of differences, ranging from 2.80 to 3.48 µM. Differences in interaction indices based on mixture formations were observed as well. For NHBE, conclusions regarding mixture interactions differed. The equimolar approach was suggestive of synergism, while the BMC-based approach suggested antagonism. Differences were observed in zebrafish as well; while both results were suggestive of antagonism, the degree differed. These results can inform future mixture studies, highlighting the importance of accounting for varying potency when investigating mixture effects.

### 4.3. Correlations of the Most Potent Chemicals Using Real-World Exposure Concentrations

A major advantage of the approach taken in Dixon, et al., 2019, is that for each wristband a unique mixture of chemicals can be identified; when all wristband data are combined, patterns of co-occurring chemical concentrations emerge (Appendix A) [2]. Correlations between the three most potent chemicals in each model system were investigated using concentration data from Dixon et al., 2019 (Figure 5). Significant pairwise relationships between potent compounds were identified, suggesting these chemicals do co-occur in real-world settings. Among the most potent chemicals in NHBE, HHCB and AHTN had the strongest correlation of 0.45, followed by BHT and HHCB and BHT and AHTN with respective correlation values of 0.34 and 0.25 (Figure 5A). These results may have implications for potential co-exposure of these compounds. HHCB and AHTN are both widely-used polycyclic musk compounds which are frequently used in scented consumer products such as perfumes, body lotions, and deodorants [88,89,90]. Additionally, these musk compounds have been identified together in consumer products and environmental media, suggesting that individuals may be exposed to these compounds as a mixture [90,91]. BHT is primarily used as an antioxidant in consumer products such as lipstick, skin lotion, deodorant, hand soap, and air fresheners [26,54,92]. HHCB has been identified at higher concentrations and more frequently in consumer products than AHTN, which may explain the stronger correlation coefficient of HHCB and BHT [89]. While we were unable to explore relationships among all three bioactive compounds in our study, studies of consumer product chemicals have identified overlap between products in which HHCB, AHTN, and BHT have been identified. Additionally, these compounds have moderate to high bioaccumulative potential. Biomonitoring studies have identified these compounds in human serum, urine, fat, and breast milk [90,92,93,94,95,96,97].

Among the compounds most active in the zebrafish model, the strongest correlation was found between DBP and DIBP (0.65), followed by BBP and DBP (0.44) and BBP and DIBP (0.38) (Figure 5B). Phthalates often occur formulated together in mixtures in body care and cosmetic products, and their frequent co-occurrence in personal passive samplers is no surprise. Co-exposure to phthalates has been investigated through measurement of urinary metabolites, with trends of higher exposure emerging for certain populations. One study investigating phthalate exposure in Beijing, China identified phthalate metabolites in every urine sample tested, with the estimated daily intake of DBP and DIBP exceeding US EPA reference values; higher metabolite levels were found among children than adults [98]. A 2014 study observed greater phthalate metabolites in urine of women and children compared to men. This trend was attributed to greater personal care product use amongst these groups, consistent with studies which identified greater phthalate metabolites in the urine of women and girls who used personal care products versus those who did not [6,99,100,101]; this suggests personal care products as a major source of phthalate exposure.

Exposure to phthalates has been associated with non-personal care product sources as well. One 2014 study detected metabolites of BBP, DBP, and DIBP in the urine of Danish children significantly correlated to the levels of parent phthalates detected in dust in the children’s homes and daycare centers [102]. The concentrations of individual urinary metabolites were significantly correlated with one another, suggesting exposure to mixtures of BBP, DBP, and DIBP, potentially through plastics around the home or components of personal care products in indoor dust.

## 5. Conclusions

This study presents a novel approach to testing real-world exposures using two alternative high-throughput screening models. For the G14 chemicals, NHBE and early life-stage zebrafish showed divergent sensitivity, each detecting bioactive chemicals that the other did not. Using this multi-system approach, we identified twelve total bioactive compounds screened across eighteen different endpoints. Additionally, the most potent compounds were screened in combination in order to investigate potential mixture effects of these commonly-occurring chemicals. Differences in mixture potency were observed depending on mixture formation approach. For investigating mixture interactions, we believe that accounting for differences in individual chemical potencies (i.e., BMC-based mixtures) is the preferred approach. A concentration addition model was used to evaluate mixture interactions. The results suggested antagonistic effects in response to both the NHBE BMC Mix and ZF BMC Mix. These mixtures were composed of chemicals representing different chemical categories. The NHBE mixture consisted of polycyclic musks (HHCB, AHTN) and a phenolic compound (BHT), while the zebrafish mixture consisted of all phthalates.

We believe this approach is particularly valuable for combining the human health relevance of human-derived in vitro models with the whole-system biological complexity of the developmental zebrafish. This study addresses an urgent need to develop novel methods to assess hazards of real-world chemical exposures quickly and accurately. Additionally, future studies could mimic this approach to further interrogate tissue-specific effects and mechanisms of chemical toxicity at the tissue level using relevant in vitro models and at the whole-organism level using early life-stage zebrafish in order to holistically capture chemical–biological interactions.

## Figures and Tables

**Figure 1 ijerph-19-03829-f001:**
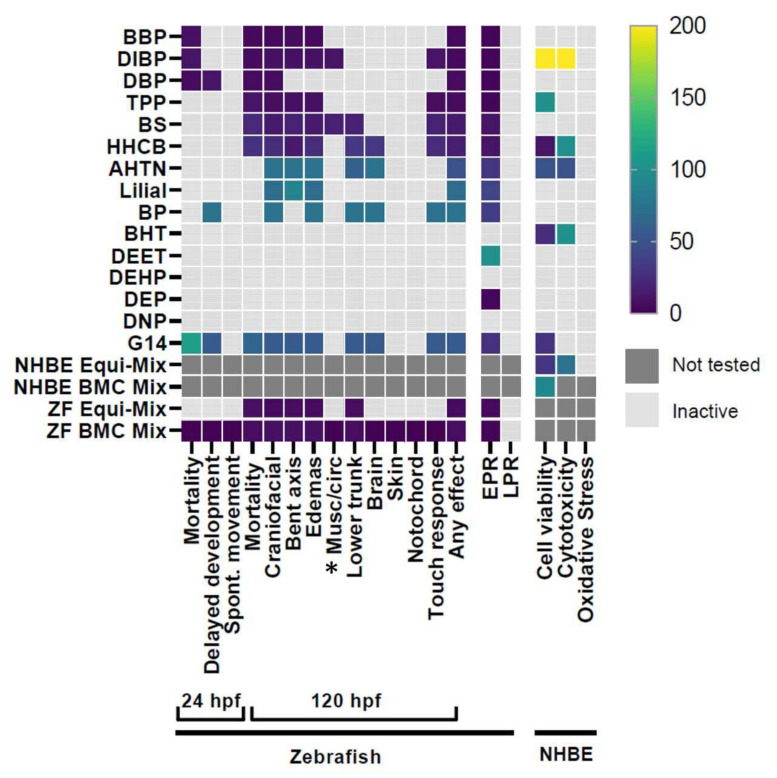
Heatmap of all endpoints across both models for each individual chemical and mixture. Colored boxes represent the lowest concentration to induce a significant effect (LEL) for each endpoint. * = Musc/circ indicates lack of circulation, malformed somites, and/or improper swim bladder formation. LEL values for each endpoint can be found in Appendix A.

**Figure 2 ijerph-19-03829-f002:**
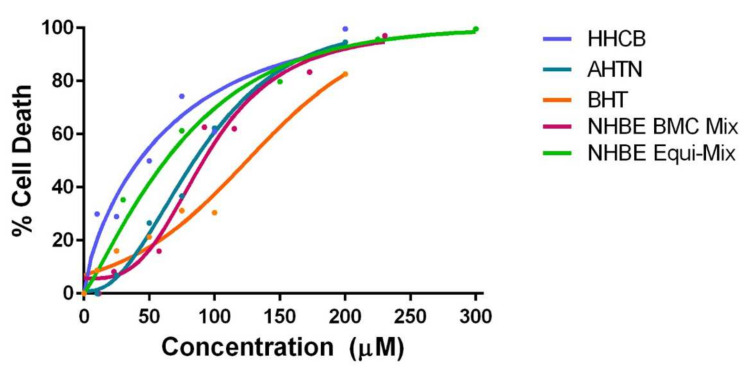
Concentration–response curves of HHCB, AHTN, BHT, BMC-anchored (NHBE BMC Mix), and equimolar (NHBE Equi-Mix) mixtures in NHBE.

**Figure 3 ijerph-19-03829-f003:**
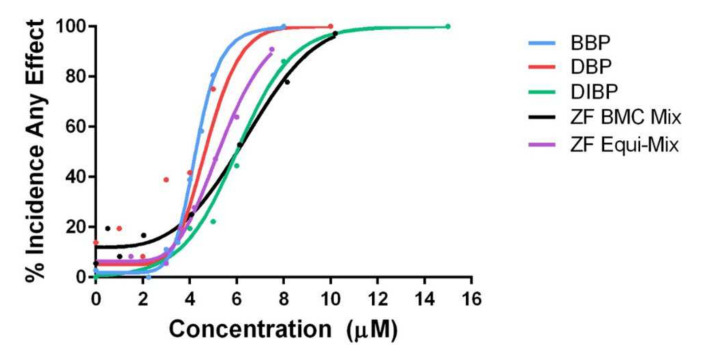
Concentration–response curves of bioactive phthalates and equimolar (ZF Equi-Mix) and BMC-anchored (ZF BMC Mix) mixtures in zebrafish.

**Figure 4 ijerph-19-03829-f004:**
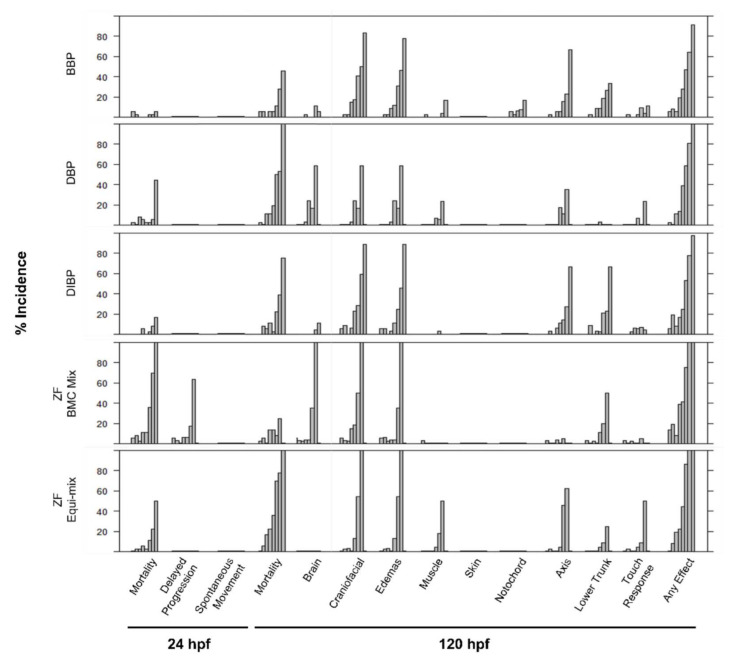
Concentration–response bar plots for incidence of each morphological endpoint from zebrafish bioactivity screening for individual phthalates and equimolar and BMC-based phthalate mixtures. Concentrations across the *y*-axis for each chemical and mixture are listed in Table 4.

**Figure 5 ijerph-19-03829-f005:**
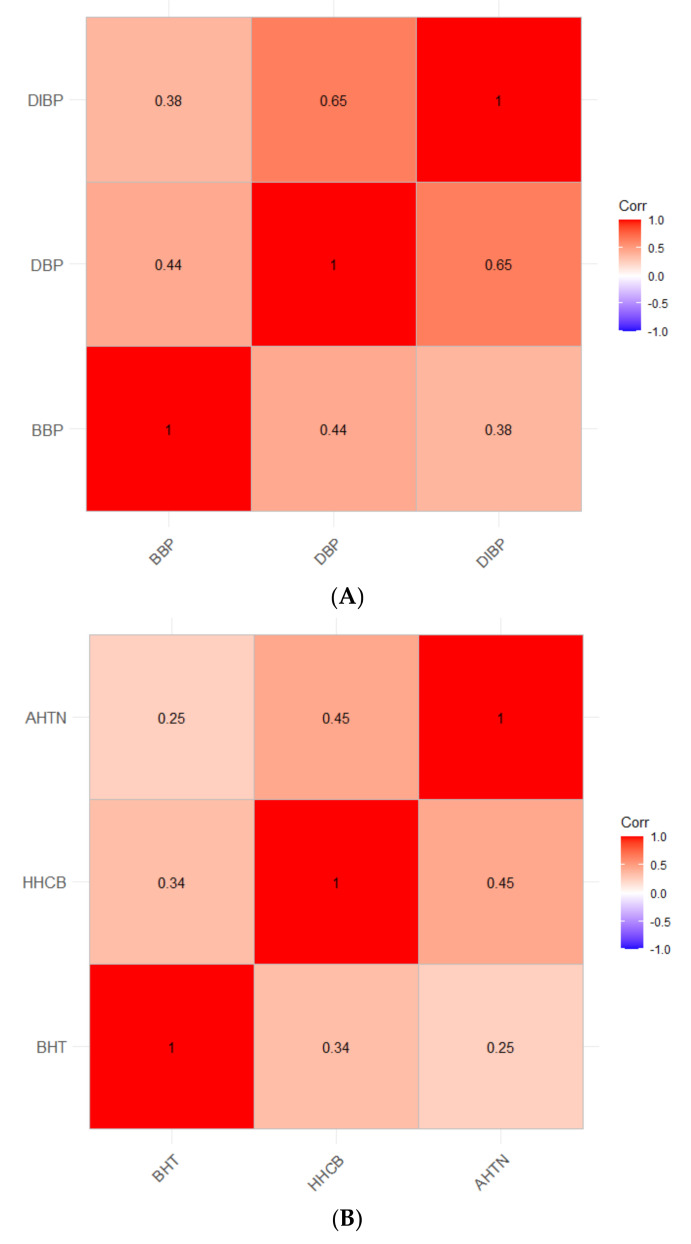
Correlation matrix investigating co-occurrence of real-world chemical concentrations for the most potent chemicals in (**A**) zebrafish and (**B**) NHBE, using exposure concentrations from Dixon et al., 2019, labeled with correlation coefficients. Only chemicals with significant correlations (*p* < 0.01) are shown.

**Table 1 ijerph-19-03829-t001:** Primary chemical structure classification and predicted functional use of each of the G14 chemicals. ** = Predicted functional use categories as determined by quantitative structure-use relationship (QSUR) models. Data was compiled and collected from the USEPA CompTox Chemicals Dashboard with data from Phillips, et al., 2017 [4]. Functional use categories with probability ≥0.3 are listed in order of highest to lowest predicted probability.

Chemical Name	Abbreviated Name	Frequency of Detection (*n* = 262)	Primary Chemical Structure Classification	Predicted Functional Use **
Benzophenone	BP	64	Benzophenone	Photoinitiator, UV absorber, crosslinker, heat stabilizer, catalyst
Benzyl butyl phthalate	BBP	66	Phthalate	Fragrance, preservative, catalyst, flavorant
Benzyl salicylate	BS	73	Salicylic acid benzyl ester	UV absorber, preservative, fragrance, antioxidant, hair dye, skin conditioner, flavorant
Bis(2-ethylhexyl) phthalate	DEHP	84	Phthalate	Fragrance, emollient, preservative, UV absorber, catalyst
Butylated hydroxytoluene	BHT	79	Phenol	Antioxidant, UV absorber, heat stabilizer, fragrance, preservative, catalyst
Diethyl phthalate	DEP	95	Phthalate	Fragrance, preservative, UV absorber, catalyst, crosslinker, skin conditioner
Diisobutyl phthalate	DIBP	85	Phthalate	Fragrance, preservative, crosslinker
Di-n-butyl phthalate	DBP	93	Phthalate	Fragrance, preservative, UV absorber, catalyst, emollient
Dinonyl phthalate	DNP	82	Phthalate	Fragrance, emollient, preservative, skin conditioner
Galaxolide	HHCB	94	Polycyclic Musk	Fragrance
Lilial	Lilial	75	Aromatic Aldehyde	Fragrance
N,N-diethyl-m-toluamide	DEET	52	Monocarboxylic acid amide	Skin protectant, catalyst, antimicrobial, colorant
Tonalide	AHTN	76	Polycyclic Musk	Fragrance
Triphenyl phosphate	TPP	52	Organophosphate	Flame retardant, catalyst, buffer

**Table 2 ijerph-19-03829-t002:** Chemical information. * = Chemicals were purchased from the listed suppliers and stock solutions were made and provided by the Oregon State University Superfund Research Center Chemical Standards Store as described above.

Chemical Name	Abbreviated Name	CAS Number	Original Supplier *	Purity (%)
Benzophenone	BP	119-61-9	Sigma Aldrich	99
Butyl benzyl phthalate	BBP	85-68-7	Sigma Aldrich	98
Benzyl salicylate	BS	118-58-1	AccuStandard	100
Bis(2-ethylhexyl) phthalate	DEHP	117-81-7	AccuStandard	99.6
Butylated hydroxytoluene	BHT	128-37-0	AccuStandard	99.8
Diethyl phthalate	DEP	84-66-2	AccuStandard	97.0
Diisobutyl phthalate	DIBP	84-69-5	AccuStandard	100
Di-n-butyl phthalate	DBP	84-74-2	CDN Isotopes	99.8
Di-n-nonyl phthalate	DNP	84-76-4	Chem Service Inc.	99.5
Galaxolide	HHCB	1222-05-5	Sigma Aldrich	87.5
Lilial	Lilial	80-54-6	Sigma Aldrich	98.4
N,N-diethyl-m-toluamide	DEET	134-62-3	AccuStandard	97.2
Tonalide	AHTN	21145-77-7	Sigma Aldrich	97.4
Triphenyl phosphate	TPP	115-86-6	Sigma Aldrich	99.8

**Table 3 ijerph-19-03829-t003:** Nominal concentrations used for NHBE bioactivity screening.

Chemical	NHBE Exposure Concentrations (µM)
* AHTN	10, 25, 50, 75, 100, 200, 400
BBP	10, 25, 50, 75, 100, 200, 400
BP	10, 25, 50, 75, 100, 200, 400
BS	10, 25, 50, 75, 100, 200, 400
* BHT	10, 25, 50, 75, 100, 200, 400
DBP	10, 25, 50, 75, 100, 200, 400
DEET	10, 25, 50, 75, 100, 200, 400
DEHP	10, 25, 50, 75, 100, 200, 400
DEP	10, 25, 50, 75, 100, 200, 400
DIBP	10, 25, 50, 75, 100, 200, 400
DNP	10, 25, 50, 75, 100, 200, 400
* HHCB	10, 25, 50, 75, 100, 200, 400
Lilial	10, 25, 50, 75, 100, 200, 400
TPP	10, 25, 50, 75, 100, 200, 400
G14 Mix	28, 70, 140, 210, 280
NHBE Equi-Mix	30, 75, 150, 225, 300
NHBE BMC Mix	11.5, 23, 57.6, 92.2, 115.2, 172.8, 230.4

* Chemicals are included in the NHBE bioactivity-based mixtures.

**Table 4 ijerph-19-03829-t004:** Definitive nominal water concentrations used for early life-stage zebrafish bioactivity screening.

Chemical	Definitive Zebrafish Exposure Concentrations (µM)
AHTN	0, 2, 10, 30, 50, 60, 75, 100
* BBP	0, 2.25, 3, 3.5, 4, 4.5, 5, 8
BP	0, 1, 2.54, 6.45, 16.4, 35, 74.8, 100
BS	0, 5, 10, 14, 18, 22, 30, 50
BHT	0, 1, 2.54, 6.45, 16.4, 35, 74.8, 100
* DBP	0, 1, 2, 3, 4, 5, 10, 20
DEET	0, 1, 2.54, 6.45, 16.4, 35, 74.8, 100
DEHP	0, 1, 3, 5, 10, 20, 40, 80
DEP	0, 1, 2.54, 6.45, 16.4, 35, 74.8, 100
* DIBP	0, 2, 4, 5, 6, 8, 10, 15
DNP	0, 2, 10, 30, 50, 70, 80, 100
HHCB	0, 10, 14, 16, 20, 24, 28, 32
Lilial	0, 20, 40, 60, 70, 80, 90, 100
TPP	0, 2, 3, 4, 5, 6, 8, 10
G14 Mix	0, 28, 56, 63, 70, 77, 84, 112
ZF BMC Mix	0, 0.51, 1.02, 2.04, 4.08, 6.13, 8.16, 10.2
ZF Equi-Mix	0, 1.5, 3, 3.6, 4.2, 5.1, 6, 7.5

* Chemicals are included in the zebrafish bioactivity-based mixtures.

**Table 5 ijerph-19-03829-t005:** Zebrafish morphology endpoints assessed at 24 and 120 h post fertilization.

Zebrafish Morphological Endpoints	
24 hpf	mortality, delayed progression, spontaneous movement
120 hpf	mortality, edemas, bent axis, touch response, and craniofacial, muscular/cardiovascular, lower trunk, brain, skin, notochord malformations

**Table 6 ijerph-19-03829-t006:** Description of each mixture screened in NHBE and zebrafish.

Mixture Name	Model	Chemical Components	Concentration Determination
G14 Mix	NHBE and Zebrafish	All G14 chemicals	Equimolar
NHBE Equi-Mix	NHBE	AHTN, BHT, HHCB	Equimolar
NHBE BMC Mix	NHBE	AHTN, BHT, HHCB	Anchored to individual BMC_50_
ZF Equi-Mix	Zebrafish	BBP, DBP, DIBP	Equimolar
ZF BMC Mix	Zebrafish	BBP, DBP, DIBP	Anchored to individual BMC_50_

**Table 7 ijerph-19-03829-t007:** BMC values for mixtures and their bioactive components across both biological models.

Model	Chemical	BMC_10_ (µM)	BMC_50_ (µM)	Regression Model
NHBE	HHCB	3.61	42.2	Gamma
NHBE	AHTN	33.5	84.9	Gamma
NHBE	BHT	45.9	103	Logistic
NHBE	G14 Mix	1.25	47.7	Log Logistic
NHBE	NHBE Equi-Mix	8.29	54.0	Weibull
NHBE	NHBE BMC Mix	47.8	94.9	Log Logistic
Zebrafish	BBP	3.29	4.28	Log Logistic
Zebrafish	DBP	2.80	4.27	Log Logistic
Zebrafish	DIBP	3.48	5.32	Logistic
Zebrafish	G14 Mix	46.2	55.4	Log Logistic
Zebrafish	ZF Equi-Mix	3.58	5.36	Gamma
Zebrafish	ZF BMC Mix	3.58	6.41	Weibull

## Data Availability

The data presented in this study are available in the main body of this manuscript or in the Appendix A.

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
