# Peer review of "A Comparative Multi-System Approach to Characterizing Bioactivity of Commonly Occurring Chemicals"

_ijerph, 2022, doi:10.3390/ijerph19073829_

Round 1
Reviewer 1 Report
Rivera et al. provide a comparison of toxicity assessments in two models using compounds prevalent in the environment, as determined via passive sampling. Compounds are not only tested alone, but also in complex mixtures, adding complexity to this study. The paper is well-written with only minor areas of edit or clarification.
Major Edits:
- It was difficult to follow which compounds were included in each mixture.
- Line 195: Include which compounds are part of NHBE Equi-Mix
- Line 201: Include which compounds are part of ZF Equi-Mix
- Line 207: Specify which compounds are in NHBE BMC and ZF BMC Mixtures.
- Lines 205-213: Consider moving this section to talk about both mixtures relevant to the NHBE cells and then both mixtures relevant to the zebrafish. This may aid in clarity.
- Table S5 should be pulled from the supplementary information. I think it would provide more information than Table 3, which could be reduced or moved to the supplementary data.
- Results as they relate to outcomes in the two models should be expanded upon. Lines 392-403 can be expanded upon, explaining the findings. It may also be helpful to separate the results discussion into 2 paragraphs for acute effects and those at 120 hpf.
- Figure 1 contains such a great amount of information, it would be helpful to make that bigger.
- Additionally, it would be helpful to either add a second scale or specify in the figure description that the high concentrations tested in the NHBE were not tested in the zebrafish.
Minor Edits:
- The introduction is lengthy and has moments of redundancies that can be reduced.
- Lines: 63-66 These lines are redundant with the previous paragraph. These can be removed, and lines 67-71 can be moved to the previous paragraph.
- Sections 1.4-1.5 The volume of this section can also be reduced, to highlight your main few points. Additionally, I would consider section 1.4 to be the key point to end and anchor the introduction before moving to the rest of the paper.
- Table 2. Please clarify: does CDN refer to CDN Isotopes?
- Section 2.2 Please clarify solvent controls for each group.
- Section 2.4.1. Please clarify survival of solvent-exposed larvae.Was survival >90% in solvent control group at 96 hpf?
- Section 2.4.2. Tale 5 referenced in this section is missing in the text.
- Section 2.4.2. Please clarify was the presence of an abnormality in any fish out of the 36 tested at each concentration sufficient to trigger a positive response in Figure 1? Or was there a threshold required?
- Line 374: I would remove the word “cytotoxicity” to avoid confusion that there was a response in the LDH assay.
- Line 381-385. Were these compounds the most potent?
- Line 405-408: Please revise or clarify. My understanding of the figure is that more than 5 compounds cause a response in the EPR assay.
- Table 6. Please include which compounds make up the mixtures. It would be helpful to include concentrations as well .
- Line 423. Please clarify what higher responses refer to. Are these compounds more potent?
- Line 438. I would assume the 2.5 uM concentration was selected based on the mortality. It would be helpful to include the LD50 values for those compounds that make up this mixture.
- Section 3.2.2. I think this discussion would be more helpful by their respective assays and reference figures.
- Line 461: Which 3 chemicals are you referencing? Are you excluding DIBP because of the high concentrations at which the effect is observed and TPP because only viability is impacted? Please clarify in this section.
- Line 461: I’m also unsure where the number 9 comes from for the effects in zebrafish. Shouldn’t there be 11 including morphological and behavioral effects?
- Line 470-471: Sentence is contradictory, please re-word.
- Line 495: While, early gonadal tissue appears in the zebrafish appears approx. 10 dpf, you could make the case that changes in protein expression targeting the mesoderm could impact later reproductive development. Rather than argue a lack of target tissue, make the case that the timepoint examined are too early to assess reproductive endpoints in the zebrafish.
- Lines 523-539: I think it’s also interesting that G14 did not impact the Musc/circ endpoint, but this was effected (and at relatively low concentrations) for 2 compounds. Did you also test this for antagonism?
- Figure 4. Please add a description of the figure and discussion to the result section. What is the x-axis?
- Section 4.3. Please clarify that these are correlations to co-exposures of these compounds. I had to read through a few times, originally thinking this was a correlation between the models.
- Figure 5. A description and discussion in the results would be appropriate.
Also both A and B are not bold.
- Lines 602-603: Clarify that these 2 mixtures are comprised of different compounds within the G14 category.
Reviewer 2 Report
The publication "A Comparative Multi-System Approach to Characterizing Bioactivity of Commonly Occurring Chemicals" is an interesting scientific study.
The entire publication is edited properly, except for an overly extensive introduction, abstract and conclusion. Many of the sentences in these parts of the manuscript are repeated. With the exception of this remark, the topic of the publication is interesting and properly developed.
Due to the applicability of the obtained results, the enormous amount of work and the use of two experimental models, I propose to accept the manuscript "A Comparative Multi-System Approach to Characterizing Bioactivity of Commonly Occurring Chemicals" in the International Journal of Environmental Research and Public Health" for printing, after considering my comment.
Reviewer 3 Report
This study describes a novel approach to broadly assess the bioactivity of chemical exposures by coupling passive chemical sampling with high-throughput toxicity screenings using in vitro and non-mammalian in vivo models. The study is innovative, but the following corrections need to be made.
- Abstract: In the abstract, the basis of the thesis is not sufficient, the author had better to reorganize the language.
- Discussion: This paper only compared the biological activities, but did not further explore the reasons why the mixture induced different biological activities.
- There are too many tables and figures in the Supplementary Material. We suggest merging related charts. For example, Table S4 has a lot of “NA” padding.
- Why NHBE cells were selected?
- Line195-204, there is no basis for selection of zebrafish exposure duration and concentration in this paper. Is it related to the environmental dose? Please add relevant description.
- In figure 1, why cytotoxicity endpoint in NHBE (BMC Mix) not be measured?
- In figure 2 and 3, it is suggested put the % after the English expression. It is suggested that the supplement to the table should be placed at the bottom.
- Line 430-431, “an interaction index of 0.93”, can you explain the specific calculation process?
- Line 435, Phthalates include BBP, DBP, and DIBP. It is suggested to modify the description.
Reviewer 4 Report
I've presented my comments in the attached file
